# RETRACTED: The Effects of Taurocholic Acid on Biliary Damage and Liver Fibrosis Are Mediated by Calcitonin-Gene-Related Peptide Signaling

**DOI:** 10.3390/cells11091591

**Published:** 2022-05-10

**Authors:** Romina Mancinelli, Ludovica Ceci, Lindsey Kennedy, Heather Francis, Vik Meadows, Lixian Chen, Guido Carpino, Konstantina Kyritsi, Nan Wu, Tianhao Zhou, Keisaku Sato, Luigi Pannarale, Shannon Glaser, Sanjukta Chakraborty, Gianfranco Alpini, Eugenio Gaudio, Paolo Onori, Antonio Franchitto

**Affiliations:** 1Department of Anatomical, Histological, Forensic Medicine and Orthopedics Sciences, Sapienza University of Rome, 00161 Rome, Italy; romina.mancinelli@uniroma1.it (R.M.); luigi.pannarale@uniroma1.it (L.P.); eugenio.gaudio@uniroma1.it (E.G.); paolo.onori@uniroma1.it (P.O.); 2Division of Gastroenterology and Hepatology, Department of Medicine, Indiana University School of Medicine, Indianapolis, IN 46202, USA; lceci@iu.edu (L.C.); linkenn@iu.edu (L.K.); heafranc@iu.edu (H.F.); vikmead@iu.edu (V.M.); chenlix@iu.edu (L.C.); kkyritsi@iu.edu (K.K.); nawu@iu.edu (N.W.); zhouv@iu.edu (T.Z.); keisato@iu.edu (K.S.); galpini@iu.edu (G.A.); 3Richard L. Roudebush VA Medical Center, Indianapolis, IN 46202, USA; 4Department of Movement, Human and Health Sciences, University of Rome “Foro Italico”, 00135 Rome, Italy; guido.carpino@uniroma4.it; 5Department of Medical Physiology, Texas A&M University, Bryan, TX 77807, USA; sglaser@tamu.edu (S.G.); schakraborty@tamu.edu (S.C.)

**Keywords:** bile acid, cAMP, biliary senescence, sensory innervation

## Abstract

Background & aims: Cholangiocytes are the target cells of liver diseases that are characterized by biliary senescence (evidenced by enhanced levels of senescence-associated secretory phenotype, SASP, e.g., TGF-β1), and liver inflammation and fibrosis accompanied by altered bile acid (BA) homeostasis. Taurocholic acid (TC) stimulates biliary hyperplasia by activation of 3′,5′-cyclic cyclic adenosine monophosphate (cAMP) signaling, thereby preventing biliary damage (caused by cholinergic/adrenergic denervation) through enhanced liver angiogenesis. Also: (i) α-calcitonin gene-related peptide (α-CGRP, which activates the calcitonin receptor-like receptor, CRLR), stimulates biliary proliferation/senescence and liver fibrosis by enhanced biliary secretion of SASPs; and (ii) knock-out of α-CGRP reduces these phenotypes by decreased cAMP levels in cholestatic models. We aimed to demonstrate that TC effects on liver phenotypes are dependent on changes in the α-CGRP/CALCRL/cAMP/PKA/ERK1/2/TGF-β1/VEGF axis. Methods: Wild-type and *α-CGRP*^−/−^ mice were fed with a control (BAC) or TC diet for 1 or 2 wk. We measured: (i) CGRP levels by both ELISA kits in serum and by *q*PCR in isolated cholangiocytes (CALCA gene for α-CGRP); (ii) CALCRL immunoreactivity by immunohistochemistry (IHC) in liver sections; (iii) liver histology, intrahepatic biliary mass, biliary senescence (by β-GAL staining and double immunofluorescence (IF) for p16/CK19), and liver fibrosis (by Red Sirius staining and double IF for collagen/CK19 in liver sections), as well as by *q*PCR for senescence markers in isolated cholangiocytes; and (iv) phosphorylation of PKA/ERK1/2, immunoreactivity of TGF-β1/TGF- βRI and angiogenic factors by IHC/immunofluorescence in liver sections and *q*PCR in isolated cholangiocytes. We measured changes in BA composition in total liver by liquid chromatography/mass spectrometry. Results: TC feeding increased CALCA expression, biliary damage, and liver inflammation and fibrosis, as well as phenotypes that were associated with enhanced immunoreactivity of the PKA/ERK1/2/TGF-β1/TGF-βRI/VEGF axis compared to BAC-fed mice and phenotypes that were reversed in α-CGRP^−/−^ mice fed TC coupled with changes in hepatic BA composition. Conclusion: Modulation of the TC/ α-CGRP/CALCRL/PKA/ERK1/2/TGF-β1/VEGF axis may be important in the management of cholangiopathies characterized by BA accumulation.

## 1. Introduction

In addition to regulating ductal bicarbonate secretion [1], cholangiocytes are the targets of cholestatic liver injury: (i) in experimental models such as extrahepatic bile duct ligation (BDL) and the multidrug gene 2 knockout (Mdr2^−/−^) mouse model of primary sclerosing cholangitis (PSC) [2,3,4]; (ii) following feeding of specific bile acids (BAs) (e.g., taurocholate (TC), taurolithocholate, and ursodeoxycholate, UDCA) [5,6,7]; and (iii) in human cholangiopathies such as PSC and primary biliary cholangitis (PBC) [8,9]. These pathologies are characterized by increased biliary senescence and changes in intrahepatic bile duct mass (IBDM), enhanced 3′,5′-cyclic cyclic adenosine monophosphate (cAMP)/protein kinase A (PKA)/extracellular signal-regulated kinase 1/2 (ERK1/2) signaling and liver fibrosis that can be triggered by paracrine mechanisms through the release of biliary senescence-associated secretory phenotypes (SASPs) such as transforming growth factor- β1 (TGF-β1) [10,11,12].

There is growing information regarding the role of BAs, which accumulate during cholestasis, in the regulation of biliary functions [6,7,13]. For example, the in vivo administration of TC to normal rats enhanced IBDM and secretin-stimulated biliary secretion through the increased activity of the apical sodium-dependent BA transporter (ASBT) [6,7]. In female Mdr2^−/−^ mice, TC has been shown to modulate liver phenotypes through increased biliary expression of ERK1/2/H19 signaling [3]. Also, TC prevents biliary damage by enhanced vascular endothelial growth factor (VEGF) expression in rat models of ischemia reperfusion of the hepatic artery [14]. There is evidence that other BAs (e.g., UDCA) and BA farnesoid X receptor (FXR) agonists (e.g., obeticholic acid) ameliorate the phenotypes of PBC and PSC [5,15,16].

Several studies have identified two afferent nerve pathways in the liver, the vagal afferent nerve and the spinal afferent nerve pathways [17], as well as an efferent function that is mediated through the release of sensory neuropeptides such as α-calcitonin gene-related peptide (α-CGRP, synthesized from the calcitonin gene transcript) [17,18] from their peripheral nerve terminals that regulate liver phenotypes [17,19]. CGRP-positive nerves are present in the fibromuscular layer of the biliary epithelium within portal areas [20]. α-CGRP is mainly synthesized in the sensory neurons of the dorsal root ganglion, which terminates peripherally on the tissues that are innervated by the sensory nervous system [21]. CGRP effects are mediated by an interaction with the calcitonin receptor-like receptor (CALCR) and the accessory protein receptor activity modifying protein 1 (RAMP-1), which is an interaction that leads to increased cAMP levels [22]. We have previously shown that: (i) α-CGRP stimulates cAMP-dependent biliary proliferation, senescence, and liver fibrosis by enhanced biliary secretion of SASPs [17,23]; (ii) knock-out of α-CGRP reduces biliary hyperplasia by decreased cAMP levels [17], which is a second messenger regulating biliary proliferation [24]; and (iii) knock-out of α-CGRP decreases biliary senescence and liver fibrosis through the decreased activation of the p38 and C-Jun N-terminal protein kinase mitogen-activated protein kinase (MAPK) signaling pathway [23]. Although there is growing information regarding the role of the central nervous system in the modulation of BA-dependent effects on liver functions, the role of sensory innervation in the modulation of BA-dependent effects on liver phenotypes in normal and cholestatic models remains unclear. Therefore, we aimed to demonstrate that the TC effects on biliary/liver phenotypes are associated with changes in the cAMP/PKA/ERK1/2 signaling, which activates the α-CGRP/TGF-β1/VEGF axis.

## 2. Materials and Methods

### 2.1. Materials

Reagents were purchased from Sigma-Aldrich Chemical Co. (St. Louis, MO, USA) unless otherwise indicated. The mRNA from isolated cholangiocytes were extracted with the mirVANA^TM^ miRNA isolation Kit (AM1561, Invitrogen). The quality and quantity of RNA were determined by a Nano Drop 2000 Spectrophotometer (Thermo Scientific, Waltham, MA, USA). The iScript cDNA Synthesis Kit and iTaq Universal SYBR Green Supermix were purchased from Bio-Rad Laboratories (Hercules, CA, USA). The mouse and human PCR primers were purchased from Qiagen (Germantown, MD, USA) (Appendix A). The TGF-β1 (sc-146), TGF-βRI (sc-101574), and VEGF-A (sc-152) antibodies were purchased from Santa Cruz Biotechnology (Santa Cruz, CA, USA). The rat monoclonal antibody against CK19 (TROMA-III) was purchased from Developmental Studies Hybridoma Bank (Iowa City, IA, USA). The antibodies against phospho-cAMP-dependent protein kinase (pPKA, ab5815), cyclin-dependent kinase inhibitor 2A (p16, ab189034), collagen I (ab34710), CD31 (ab28364), and TGF-β1 (ab92486) were purchased from Abcam (Cambridge, MA, USA). CRLR/CGRPR1 (bs-1860R) rabbit polyclonal antibody was purchased from Bioss Antibody (Woburn, MA, USA). The Phospho-p44/42 MAPK (*p*ERK1/2) rabbit monoclonal antibody (137F5) was purchased from Cell Signaling Technology, Inc. (Danvers, MA, USA). The specific secondary antibodies for immunofluorescence (Cy2 anti-rabbit and Cy3 anti-rat) were purchased from Jackson Immunochemicals (West Grove, PA, USA).

### 2.2. Animal Models

All animal experiments were performed in accordance with protocols approved by either the Baylor Scott & White Health or Indiana University School of Medicine IACUC Committee. The C57BL/6 wild-type (WT) mice were purchased from Charles River (Wilmington, MA, USA). The α-CGRP knockout (*α-CGRP*^−/−^) model, which is generated as described [25], is established in our breeding colony and has been used in previous studies [17,23]. The animals were kept in a temperature-controlled environment (20–22 °C) with 12-h light/dark cycles with free access to drinking water and fed ad libitum standard mouse chow. Studies were performed in male WT and α-CGRP^−/−^ mice (12 wk of age, 20–25 g) that were fed an ad libitum BA control diet (BAC, AIN 76) or 1% taurocholic acid diet (TC, representing an approximate dose of 275 μmol/day) for 1 and 2 wk [6,7] with free access to drinking water; diets were purchased from Dyets Inc. (Bethlehem, PA, USA). Before collection of specimens, the animals were anesthetized with euthasol (50–90 mg/kg BW). We collected total liver and cholangiocytes from all groups, which were stored at −80 °C before use. The body weight, liver weight, and liver/body weight ratio were measured to evaluate liver damage (Table 1).

### 2.3. Isolated Cholangiocytes

The isolation of mouse cholangiocytes was performed by immunoaffinity separation [26] using a monoclonal antibody rat IgG_2a_ (a gift from Dr. R. Faris, Brown University, Providence, RI, USA) against an unidentified antigen expressed by all intrahepatic cholangiocytes [27].

### 2.4. Measurement of CGRP in Serum and Immunoreactivity of CALCRL in Liver Sections

The CGRP immunoreactivity of α-CGRP receptor components (CALCRL) was analyzed by immunohistochemistry in paraffin-embedded liver sections (4 μm thick). A digital scanner (Aperio Scanscope CS System; Aperio Digital Pathology, Leica Biosystems, Milan, Italy) was used to scan the slides; the analysis was performed by an ImageScope 12.3.3 (Leica Biosystems) for each experimental group (n = 7 animals per groups; five different areas were analyzed from each group of mice). We performed *q*PCR for CALCA (gene for α-CGRP) in isolated cholangiocytes from WT mice fed the BAC or TC diet for 1 and 2 wk.

### 2.5. Evaluation of Liver Damage

All paraffin-embedded liver sections (4 μm thick) were stained with hematoxylin and eosin (H&E) and were analyzed by a pathologist in a coded fashion. The sections were examined with a light microscope (Leica Microsystems DM 4500; Wetzlar, Germany), imaged with the Jenoptik ProgRes C10 Plus (Jena, Germany), and analyzed with the Image Analysis System (Delta Sistemi, Rome, Italy).

### 2.6. Evaluation of Intrahepatic Bile Duct Mass (IBDM)

IBDM was measured by semiquantitative immunohistochemistry in paraffin-embedded liver sections (4 µm, thick) as the area occupied by % CK19-positive bile ducts over total liver area. Negative controls with the omission of the primary antibody (replaced with normal serum from the same species) were included for all samples. The slides were scanned by digital scanner (Aperio Scanscope CS System; Aperio Digital Pathology, Leica Biosystems) and a semiquantitative analysis of the positive staining was performed for each experimental group (n = 3 animals per group) and processed by ImageScope 12.3.3 (Leica Biosystems).

### 2.7. Evaluation of Biliary Senescence

Biliary senescence was evaluated by staining for SA-β-galactosidase (SA-β-GAL) using commercially available kits (MilliporeSigma, Billerica, MA, USA) in frozen liver sections (10 µm thick); a semiquantitative analysis of the positive staining was performed for each group of mice (n = 3 animals per group; we analyzed five different areas from each group of mice). The slides were scanned by a digital scanner (Aperio Scanscope CS System; Aperio Digital Pathology, Leica Biosystems) and evaluated with ImageScope 12.3.3 (Leica Biosystems). Biliary senescence was evaluated by immunofluorescence for p16 in frozen liver sections (6 µm thick, co-stained with CK19) from the selected experimental groups. For all immunoreactions, negative controls (the primary antibody was replaced with bovine-serum-albumin) were performed; pre-immune serum was also added. Immunofluorescent staining was analyzed with an SP8 confocal microscope platform from Leica Microsystems (Leica Microsystems Inc., Buffalo Grove, IL, USA). To further support the immunohistochemical analysis, we measured the mRNA expression of p21 and p16 in isolated cholangiocytes from the experimental groups of mice by *q*PCR.

### 2.8. Evaluation of Collagen Content

Hepatic fibrosis was determined by Fast Green/Sirius Red staining in paraffin-embedded liver sections (4 μm thick) and evaluated with a Leica Microsystems DM 4500 B Microscope (Weltzlar, Germany) equipped with a JenoptikProg Res C10 Plus Videocam (Jena, Germany). Liver fibrosis was also measured by immunofluorescence for collagen-1 in frozen liver sections (6 μm thick, co-stained with CK19) and analyzed by the SP8 confocal microscope platform from Leica Microsystems (Leica Microsystems Inc., Buffalo Grove, IL, USA).

### 2.9. Measurement of cAMP/pERK/VEGF Signaling, Hepatic BA Composition, and Biliary Immunoreactivity/Expression of TGF-β1 and TGF-βRI

The next sets of experiments were based on the background that: (i) TC increases biliary hyperplasia by activation of cAMP signaling [6,7]; (ii) cAMP/PKA/ERK1/2 signaling is a key regulator of biliary proliferation [23]; (iii) α-CGRP/CALCRL signaling is regulated by cAMP-dependent ERK1/2 phosphorylation [17,28]; (iv) α-CGRP/CALCRL regulates VEGF expression [29]; and (v) inhibition of α-CGRP decreases TGF-β1 expression [23]. We performed Ingenuity Pathway Analysis (IPA) [4] software version 01-16 (Qiagen, Redwood City, CA) that suggested a link between TC and cAMP/α-CGRP/VEGF//TGF-β1 signaling (see Section 3, Figure 6). IPA is a web-based functional analysis software that allows us to search for targeted information on genes, proteins, chemicals, diseases, and drugs, as well as building custom biological models in life science research. In our mouse models, we evaluated: (i) changes in BA composition (which may partly explain the effects of TC on liver phenotypes) in total liver samples from three mice for each experimental group with a Shimadzu liquid chromatography/mass spectrometry (LC-MS) 8600 system at the McGuire Research Institute, Richmond, VA, USA [30]; (ii) phospho-PKA by immunohistochemistry in paraffin-embedded liver sections (4 μm thick); and (iii) *p*ERK1/2 by immunofluorescence in frozen liver section (6 μm thick)—all the stainings were performed in n = 3 animals per group.

We measured the immunoreactivity of TGF-β1 and TGF-βRI in paraffin-embedded liver sections (4 µm thick). Following staining, the slides were scanned by a digital scanner (Aperio Scanscope CS System, Aperio Digital Pathology, Leica Biosystems, Milan, Italy) and processed by ImageScope. Moreover, an image analysis algorithm was used to count the percentage of immunostained areas in at least 6 non-overlapping fields at 10× to quantify the presence of TGF-β1- and TGF-βRI-positive cholangiocytes. We also measured the mRNA expression of TGF-β1 and TGF-β1RI in isolated cholangiocytes by *q*PCR, and we co-stained CK19 with TGF-β1 to evaluate its relationship with cholangiocytes.

VEGF-A and CD31 (marker of endothelial cells) [31] were measured by immunohistochemistry in paraffin-embedded liver sections (4 μm thick) and immunofluorescence in frozen liver sections (6 μm thick), respectively. Immunohistochemical staining was analyzed with a digital scanner (Aperio Scanscope CS System; Aperio Digital Pathology, Leica Biosystems) and immunofluorescent staining was evaluated with an SP8 confocal microscope platform from Leica Microsystems (Leica Microsystems Inc., Buffalo Grove, IL, USA).

### 2.10. Statistical Analysis

The data are presented as the mean ± SEM (standard error). Bile acids data are presented as the mean ± standard deviation. One-way ANOVA was used when more than two groups were analyzed, followed by an appropriate post hoc test with GraphPad Prism 9 (San Diego, CA, USA). A value of *p* < 0.05 was considered significant.

## 3. Results

### 3.1. Measurement of CGRP Serum Levels and Immunoreactivity of CALCRL in Liver Sections

The mRNA expression of CALCA (gene for *α-CGRP)* is significantly increased in isolated cholangiocytes of TC-fed-WT mice compared to BAC-fed WT mice (Figure 1A). There are two different forms of calcitonin gene-related peptides (α and β forms), which are structurally similar and differ only in two amino acids in rats and three amino acids in humans [32]. Based on the data related to the CGRP serum levels and CALCA gene expression, we suggest that: (i) *α-CGRP*^−/−^ mice did not show complete absence of α-CGRP neuropeptide in serum due to cross-reactivity between α-CGRP and β-CGRP detected by the ELISA kit; and (ii) TC modulates the expression of the CALCA gene that is likely in cholangiocytes. We demonstrated that the biliary immunoreactivity of CALCRL increased in both *α-CGRP*^−/−^ mice fed the BAC or TC diet compared to TC-fed WT mice (Figure 1B), which was likely due to a compensatory mechanism that leads to the enhanced expression α-CGRP receptor components in the absence of its substrate (α-CGRP). These findings are supported by a previous study showing enhanced mRNA expression of the CGRP-receptor components (*CRLR*, *RAMP-1,* and *RCP*) in cholangiocytes from normal and BDL *α-CGRP^−/−^* compared to WT mice [23].

### 3.2. Evaluation of Liver Damage

No changes in liver inflammation, necrosis, and lobular damage were observed between WT and *α-CGRP*^−/−^ mice treated with BAC or TC for 1 wk (Figure 2). We observed lobular inflammation consisting of infiltration of polymorphonuclear cells (yellow arrows, Figure 2) and lobular damage in WT mice fed TC for 2 wk compared to WT mice fed the BAC or TC diet for 1 wk. Furthermore, in WT mice fed the TC diet for 2 wk, we observed indued swelling, ballooning, and damage of the hepatocytes (green arrows, Figure 2). These phenotypes were reduced in *α-CGRP**^−/−^* mice fed TC for 2 wk. Furthermore, *α-CGRP**^−/−^*mice fed TC for 2 wk showed lower inflammatory infiltration (Figure 2). WT mice fed TC diet for 2 wk displayed a significant increase in their liver-to-body weight ratio (LW/BW, an index of liver cell proliferation) [33] compared to WT mice fed the BAC diet for 2 wk; however, there were no significant changes in liver weight, body weight, and liver-to-body weight ratio between the other experimental groups (Table 1).

Similar to a previous study [6], the feeding of WT mice with TC (for 1 and 2 wk) induces a significant increase in IBDM compared to BAC-fed WT mice (Figure 3). TC did not increase IBDM in *α-CGRP*^−/−^ compared to *α-*CGRP^−/−^ mice fed BAC (Figure 3), and no significant difference in IBDM was observed between WT and *α-CGRP*^−/−^ mice fed BAC (Figure 3).

In WT mice fed TC, there was an increase in biliary senescence compared to BAC-fed WT mice, as observed by staining for SA-β-GAL (blue arrows, Figure 4A) and p16 (Figure 4B). However, TC did not significantly increase biliary senescence in *α-CGRP*^−/−^ mice compared to *α-CGRP*^−/−^ mice fed BAC (Figure 4); no significant difference in biliary senescence was observed between WT and *α-CGRP*^−/−^ mice treated with BAC (Figure 4). These phenotypes were further supported by *q*PCR for p16 and p21 in isolated cholangiocytes from the selected experimental groups (Figure 4C).

### 3.3. Evaluation of Collagen Deposition

By Fast Green/Sirius Red staining, WT mice fed TC for 1 and 2 wk showed higher collagen deposition compared to WT mice fed BAC (Figure 5A). Furthermore, through immunofluorescence we found that TC feeding did not induce a significant increase in collagen deposition in α-*CGRP*^−/−^ mice compared to *α-CGRP*^−/−^ mice fed BAC; no significant changes in liver fibrosis were observed between normal WT and *α-CGRP*^−/−^ mice treated with BAC (Figure 5B).

### 3.4. Measurement of Hepatic BA Composition, cAMP/pERK/VEGF Signaling, Hepatic BA Composition, and Biliary Immunoreactivity/Expression of TGF-β1, mdnTGF-βRI, and Angiogenic factors

According to previous studies [6,7,17,29], and supported by Ingenuity Pathway Analysis (IPA, Figure 6), TC modulates the expression of the α-CGRP/CALCRL axis, TGF-β1 signaling, and liver angiogenic factor expression by cAMP/ERK signaling-dependent pathways. According to the hepatic BA composition analysis, WT mice fed TC for 1 or 2 wk displayed decreased levels of taurohyodeoxycholic acid (THDCA), ω-muricholic acid (ω-MCA), β-muricholic acid (β-MCA), taurochenodeoxycholic acid (TCDCA), chenodeoxycholic acid (CDCA), and lithocholic acid (LCA) compared to WT mice fed BAC for 1 or 2 wk (Appendix A). Furthermore, *α-CGRP*^−/−^ mice fed TC for 1 or 2 wk showed a reduction of β-MCA, TCDCA, CDCA, and LCA compared to WT mice fed BAC for 1 or 2 wk (Appendix A). WT mice fed the TC diet for 2 wk displayed decreased levels of total primary unconjugated BAs, total secondary unconjugated BAs, and increased levels of the ratio of total conjugated BA to total unconjugated BA when compared to WT mice fed BAC for 2 wk. On the other hand, *α-CGRP*^−/−^ mice fed TC for 1 wk showed an increased ratio of total primary conjugated BA to total primary unconjugated BA and an increase in the ratio of total conjugated BA to total unconjugated BA when compared with WT mice fed TC for 1 wk (Appendix A). Overall, the data show that: (i) there are no changes in BA composition in both WT and *α-CGRP*^−/−^ mice; (ii) TC feeding reduces the levels of β-MCA, TCDCA, CDCA, and LCA in both WT and *α-CGRP*^−/−^ mice; and (iii) WT and *α-CGRP*^−/−^ mice fed TC for 2 wk showed a reduced total conjugated to total unconjugated BA ratio. BA profiling also indicated that TC levels are reduced in *α-CGRP*^−/−^ mice fed TC for 2 wk, which is consistent with reduced cholangiocyte proliferation [6,13] (Appendix A); however, the increase in TC levels (observed after 1 wk TC feeding of *α-CGRP*^−/−^ mice) may be due to a compensatory mechanism due to changes in BA composition. BA profiling showed that the increase in the total conjugated primary BA pool parallels the increase in biliary proliferation in WT mice fed TC (Appendix A).

We demonstrated enhanced biliary pPKA immunoreactivity in liver sections from TC-fed WT mice compared to BAC-fed WT mice; these phenotypes were reversed in α-*CGRP*^−/−^ mice fed TC compared to *α-CGRP*^−/−^ mice fed BAC (Figure 7A). Moreover, pERK immunoreactivity (mostly expressed in the nuclei of cholangiocytes (Figure 7B)) was increased in TC-fed WT mice compared to BAC-fed WT mice, which was significantly decreased in α-*CGRP*^−/−^ mice fed TC compared to *α-CGRP*^−/−^ mice fed BAC (Figure 7B). TC-fed WT mice displayed an elevated expression of TGF-β1 and TGF-βRI in liver sections compared to WT mice fed BAC (Figure 8 and Figure 9), which was reduced in *α-CGRP*^−/−^ mice fed TC. Lastly, WT mice fed TC showed an enhanced immunoreactivity to CD31 and VEGF-A in liver sections, which was unchanged in *α-CGRP*^−/−^ mice fed TC compared to *α-CGRP*^−/−^ mice fed BAC (Figure 10A,B).

Several studies have demonstrated the role of TC in the modulation of biliary functions from changes in different transduction pathways such as cAMP and Ca^2+^-dependent protein kinase C [14,34]. In this regard, the second messenger cAMP is a key modulator of biliary proliferation, and its levels are influenced by neurotransmitters, gastrointestinal hormones, and BAs including TC [6,7]. For example, TC stimulates biliary proliferation and secretin receptor expression as well as secretin-stimulated bicarbonate-rich ductal secretion by enhanced biliary cAMP levels [6,35]. Supporting the trophic effect of TC on biliary functions, the TC stimulated cystic fibrosis transmembrane conductance activity is regulated by a mechanism involving cAMP-dependent, ASBT-mediated bile salt absorption [36]. Further, TC triggers liver fibrosis in the *Mdr2^−/−^* mouse model of PSC by the activation of ERK1/2 signaling, which are phenotypes that were reduced by the downregulation of the long, non-coding RNA H19 [3].

Biliary senescence is a hallmark of PSC, PBC, nonalcoholic fatty liver disease, and liver cirrhosis, which are characterized by changes in the levels/composition of BAs [3,4,23,37,38,39]. There is growing information regarding the role of BAs in cellular senescence in chronic cholestatic liver diseases. For example, glycochenodeoxycholic acid promotes biliary autophagy in PBC (through increased mitochondrial antigens) as well as biliary senescence [40]. The possible effects of BAs on TGF-β1 signaling is also supported by a study showing that taurochenodeoxycholic acid stimulates the release of TGF-β1 from cultured Kupffer cells [41]. Consistent with the concept that BAs modulate cellular senescence, a study has shown that tauroursodeoxycholic acid decreased cellular senescence by reducing p53 and p21 expression [42]. Furthermore, the secondary BA, deoxycholic acid, increased cellular senescence, thereby decreasing the proliferation of LX2 cells (human HSC line) by a mechanism mediated by SASPs, which includes interleukin 8 and TGF-β1 [43]. In addition, UDCA inhibits fibrogenesis both in in vivo models and in vitro models in LX2 cells through the inhibition of TGF-β1-induced autophagy; however, no direct evidence exists for the role of TC in biliary senescence that may be mediated by the increased biliary expression of TGF-β1 [44].

Several studies evaluated the BA composition in the plasma, urine, and liver of different species [45,46]. It was demonstrated that mice have more BA amidation than rats [45]. As well, rats showed higher concentrations of cholic acid in their serum compared to mice (50% compared to 24%, respectively) [46]. However, no significant difference was observed in the hepatic BA composition of mouse and rats [45,46].

Liver functions are regulated by efferent, afferent, and sensory (e.g., substance P and CGRP) fibers. Regarding sensory innervation, studies have shown that α-CGRP triggers biliary hyperplasia, senescence, and liver fibrosis, and that the knock-out of α-CGRP ameliorates these liver phenotypes in cholestatic mice [17,23]. Our current data support the role of sensory innervation in the modulation of liver phenotypes during cholestasis and suggest a potential common signaling pathway (cAMP/ERK1/2/TGF-β1) between sensory innervation and BAs in the prevention of biliary damage. To support this common transduction pathway, CGRP prevents cellular senescence induced by TGF-β1 through increased klotho expression in cardiac fibroblasts [47], and the link between TC and α-CGRP/CALCRL/cAMP/ERK1/2/TGF-β1/VEGF signaling is suggested by IPA (Figure 6).

We next evaluated the effects of the TC/α-CGRP/CALCRL signaling axis on liver angiogenic factor expression by measuring the immunoreactivity of VEGF-A (a trophic biliary factor) [48] and the endothelial marker, CD31 [48,49,50], which are two angiogenic factors that are upregulated during biliary damage and that play important roles in the progression of liver phenotypes such as fibrosis [4,10]. Indeed, during biliary damage, there is proliferation and angiogenesis of the peribiliary plexus and secretion of VEGF-A from both cholangiocytes and the peribiliary plexus that support the expansion of the proliferating biliary epithelium [51,52]. Supporting our proposed signaling axis, TC maintains the balance between biliary growth and loss in cholangiopathies sustaining the expression of angiogenic factors such as VEGF [14,53]. Also, CGRP has been shown to promote ischemia-induced angiogenesis by increasing the expression of CD31, VEGF-A, and TGF-β that supports the interaction between sensory innervation and angiogenesis axes [54]. Indeed, our data confirmed that in WT mice fed TC there was increased expression of CD31 (endothelial marker) [48,49,50] and VEGF-A (pro-angiogenic factor) [48], which was reduced in *α-CGRP*^−/−^ mice fed TC.

We next evaluated the mechanisms by which TC and α-CGRP interact with each other and we propose that the stimulatory effects of the TC/α-CGRP axis on the cAMP signaling axis may be due to a number of factors, such as the weak interaction of TC or unconjugated secondary BAs (changed in our model) with Takeda G-protein-coupled receptor 5 or ASBT [7]. We also speculate that TC triggers the biliary release of α-CGRP by a cAMP/ERK1/2 pathway acting as an autocrine factor that, through binding with CALCRL, promotes biliary proliferation/senescence and liver fibrosis. Since the combination of TC feeding and the consequent CALCRL activation increases CGRP serum levels by a cAMP-dependent p-PKA and p-ERK, and since α-CGRP sustains biliary growth during BDL through the activation of ERK1/2 [55]—whereas a lack of α-CGRP/CALCRL signaling inhibits the stimulatory effects of TC on liver phenotypes—we propose that the cAMP-dependent α-CGRP/CALCRL signaling axis is a common transduction pathway mediating the effects of TC on liver phenotypes in WT and *α-CGRP*^−/−^ mice. According to IPA, TC may induce the binding of α-CGRP with its CALCRL receptor that regulates cellular events such as fibrogenesis (see our proposed working model depicted in the graphical abstract of Figure 11). Moreover, we demonstrated the novel role of TC/α-CGRP/CALCRL signaling in the homeostasis of the biliary epithelium during biliary injury, which may be due to changes in the cAMP/ERK1/2/TGF-β1/VEGF signaling axis. Based on these findings, we propose that the effects of TC on biliary/liver phenotypes are associated with correlative changes in the expression of the PKA/ERK1/2/TGF-β1/TGF-βRI/VEGF axis through changes in BA homeostasis.

## Figures and Tables

**Figure 1 cells-11-01591-f001:** CALCA gene and CALCRL expression. (**A**) TC-fed WT mice showed elevated CALCA expression (gene for α-CGRP) compared to BAC-fed WT mice. Evaluations from three cumulative preparations of cholangiocytes from five mice per group. * < *p* value vs. WT + BAC. (**B**) CALCRL (α-CGRP receptor component) staining by immunohistochemistry in liver sections. Original magnification 40×; black arrow indicates bile ducts. Data for IHC quantification are mean ± SEM of scanning slides from n = 6 different animals per groups. * < *p* value vs. WT + BAC; † < *p* value vs. WT + TC.

**Figure 2 cells-11-01591-f002:** Evaluation of liver histology infiltrates by H&E staining in liver sections from all experimental groups of mice. H&E staining showed lobular inflammation consisted of infiltration of polymorphonuclear cells (yellow arrows) in WT mice fed the TC diet for 2 wk compared to WT mice fed the BAC diet and WT mice-fed the TC diet for 1 wk. In addition, the TC diet for 2 wk in WT mice induced swelling, ballooning, and damage of hepatocytes (green arrows). The phenotypes were ameliorated in *α-CGRP^−/−^* mice fed the TC diet for 2 wk compared to WT mice treated with the TC diet for 2 wk. Original magnification 20×; scale bar 50 μm. Data are analyzed from n = 6 animals per each experimental group. Data are reported in Table 1.

**Figure 3 cells-11-01591-f003:** Knockout of α-CGRP decreases IBDM in mice fed with TC diet. In WT mice fed TC, there was a significant increase in IBDM compared to BAC-fed WT mice. TC did not increase IBDM in *α-CGRP*^−/−^ compared to *α-CGRP*^−/−^ mice fed BAC; no significant difference in IBDM was observed between normal WT and *α-CGRP*^−/−^ treated with BAC. Black arrows show bile ducts. Original magnification 10×; scale bar 100 μm Data are mean ± SEM of slides (completed scanned) from n = 3 different animals per groups. * < *p* value vs. WT + BAC; † < *p* value vs. WT + TC.

**Figure 4 cells-11-01591-f004:** Biliary senescence is decreased in TC-fed *α-CGRP*^−/−^ mice. (**A**) By SA-β-GAL staining and (**B**) immunofluorescence for p16 (co-stained for CK19, original magnification 20×; scale bar 100 μm, white box: original magnification 80×). WT TC-fed mice displayed elevated biliary senescence compared to WT-fed BAC. TC feeding did not significantly increase biliary senescence in *α-CGRP*^−/−^ mice compared to α-CGRP^−/−^ mice fed BAC diet; no significant difference in biliary senescence was observed between WT and *α-CGRP*^−/−^ mice fed BAC diet. Blue arrows show SA-β-GAL positivity in bile ducts. Original magnification 20×; scale bar 100 μm. Black box: 40×. Data are mean ± SEM of scanning slides from n = 3 different animals per groups. (**C**) mRNA expression of p21 and p16 in isolated cholangiocytes is significantly elevated in TC-fed WT mice, which is reduced in TC-fed *α-CGRP*^−/−^ mice. Data are mean ± SEM of three evaluations from three cumulative preparations of cholangiocytes from five mice per groups. ***** < *p* value vs. WT + BAC; † < *p* value vs. WT + TC.

**Figure 5 cells-11-01591-f005:** Lack of α-CGRP reduce liver fibrosis in mice fed with TC. (**A**) By Fast Sirius red staining, WT mice treated with TC for 1 or 2 wk showed elevated collagen deposition compared to WT mice treated with BAC, which was significantly reduced in *α-CGRP*^−/−^ mice fed TC. Original magnification 10X. Scale Bar 100 μm. Data are mean ± SEM from n = 6 different animals. * < *p* value vs. WT + BAC; † < *p* value vs. WT + TC. (**B**) Immunofluorescence staining shows that TC increased the immunoreactivity of collagen (green) in WT mice. *α-CGRP*^−/−^ mice fed TC diet have reduced expression of collagen compared to WT-fed TC mice. No significant changes in collagen deposition were observed between normal WT and *α-CGRP*^−/−^ mice fed BAC diet. Original magnification 20×; scale bar 100 μm.

**Figure 6 cells-11-01591-f006:** Ingenuity Pathway Analysis (IPA) shows the relationship between TC and α-CGRP/CALCRL axis and TGF-β1 signaling leading to increase liver angiogenesis by cAMP/ERK signaling-dependent pathways.

**Figure 7 cells-11-01591-f007:** Phospho-PKA and phospho-ERK immunoreactivity is reduced in *α-CGRP^−/−^* mice fed TC diet. (**A**) Immunohistochemistry of phospho-PKA in paraffin-embedded liver sections, and (**B**) immunofluorescence for phospho-ERK (green) co-stained with CK19 (red, marker of cholangiocytes, original magnification 40×; scale Bar 50 μm) in frozen liver sections from the selected groups of mice. Biliary immunoreactivity for phospho-PKA and phospho-ERK increased in liver sections from TC-fed WT mice compared to BAC-fed WT mice, which are phenotypes that were significantly decreased in *α-CGRP*^−/−^ mice fed TC compared to *α-CGRP*^−/−^ mice fed BAC. Original magnification 20×; scale bar 100 μm. Black box: original magnification 40×, scale bar 50 μm.

**Figure 8 cells-11-01591-f008:** *α-CGRP^−/−^* mice fed with TC have decreased expression of TGF-β1 in liver. (**A**) *q*PCR in isolated cholangiocytes as well as (**B**) immunohistochemistry of WT mice fed TC have increased biliary TGF-β1, which is significantly reduced in *α-CGRP*^−/−^ mice fed with TC compared to WT fed BAC. Original magnification 20×. Scale Bar 100 μm. Data are mean ± SEM of scanning slides from n = 3 different animals per groups. This phenotype was confirmed by (**C**) immunofluorescence staining of TGF-β1 co-staining with CK19. Data from *q*PCR are mean ± SEM of three evaluations from three cumulative preparations of cholangiocytes from five mice per group. Original magnification 20×. Scale Bar 20 μm.* < *p* value vs. WT + BAC. † < *p* value vs. WT + TC.

**Figure 9 cells-11-01591-f009:** TGF-βRI immunoreactivity is reduced in TC-fed *α-CGRP*^−/−^ mice. (**A**) mRNA expression of TGF-βRI is reduced in TC-fed *α-CGRP*^−/−^ mice compared to WT fed TC in isolated cholangiocytes. (**B**) Immunoreactivity of TGF-βRI was significantly increased in WT mice fed TC (compared to WT mice fed BAC), which was reduced in TC-fed *α-CGRP*^−/−^ mice compared to WT fed TC. Original magnification 20×. Scale Bar 100 μm. Black box: original magnification 40×. Data are mean ± SEM of scanning slides from n = 3 different animals per groups. Data are mean ± SEM of three evaluations from three cumulative preparations of cholangiocytes from five mice per group. ***** < *p* value vs. WT + BAC; † < *p* value vs. WT + TC.

**Figure 10 cells-11-01591-f010:** Knockout of α-CGRP decreased angiogenic factor expression in mice fed with TC diet. (**A**) TC treated WT mice showed enhanced immunoreactivity of CD31 (angiogenic marker) compared to WT mice treated with BAC diet; this parameter is decreased in TC-fed *α-CGRP*^−/−^ mice. Original Magnification 20×; scale bar 100 μm. (**B**) Immunohistochemistry for VEGF-A in liver section demonstrated reduction of the angiogenic factor both in BAC- and TC-fed *α-CGRP*^−/−^ mice. Original magnification 20×; scale bar 100 μm. Data are mean ± SEM of scanning slides from n = 3 animals per each experimental group.

**Figure 11 cells-11-01591-f011:** Working model illustrating the effects of TC on liver phenotypes in WT and *α-CGRP*^−/−^ mice. (**A**) TC effects on liver phenotypes are due to changes in hepatic BA composition leading to enhanced expression of the cAMP/PKA/ERK signaling pathway. As a consequence, cholangiocytes acquire a neuroendocrine phenotype and promote the release of α-CGRP neuropeptide. By an autocrine mechanism, α-CGRP binds CGRP-receptor components (CALRL, RAMP-1, and RCP) that are expressed on cholangiocytes and promotes cholangiocyte proliferation, liver fibrosis, and angiogenesis. (**B**) In the absence of α-CGRP neuropeptide, cholangiocytes overexpress CGRP-receptor components (CALRL, RAMP-1, and RCP) as a compensatory mechanism and reduce biliary proliferation, biliary senescence, and liver fibrosis and angiogenesis. Graphic illustration was created by BioRender.com.

**Table 1 cells-11-01591-t001:** Assessment of liver weight, body weight, and liver-to-body weight ratio. Evaluation of IBDM and Biliary Senescence.

**Treatment**	**WT+** **BAC 1 wk**	**WT+** **TC 1 wk**	**α** **-CGRP−/±** **BAC 1 wk**	**α** **-CGRP−/±** **TC 1 wk**	**WT+** **BAC 2 wk**	**WT+** **TC 2 wk**	**α** **-CGRP−/±** **BAC 2 wk**	**α** **-CGRP−/±** **TC 2 wk**
Liver weight (g)	1.4 ± 0.1n = 8	1.5 ± 0.1n = 17	1.6 ± 0.1n = 9	1.4 ± 0.1n = 10	1.3 ± 0.1n = 10	1.6 ± 0.1n = 11	1.4 ± 0.01n = 7	1.3 ± 0.1n = 14
Body weight (g)	27.7 ± 1.6n = 8	24.8 ± 0.5n = 17	25.4 ± 1.4n = 9	22 ± 1.1n = 10	27.6 ± 1.0n = 10	25.0 ± 0.9n = 11	26.8 ± 1.1n = 7	23.4 ± 1.1n = 14
LW/BW (%)	5.1 ± 0.3n = 8	6.2 ± 0.2n = 17	6.50 ± 0.7n = 9	5.9 ± 0.3n = 10	4.7 ± 0.3n = 10	6.7 ± 0.6 *n = 11	5.3 ± 0.3n = 7	5.7 ± 0.2n = 14
Inflammatory infiltrated area	0.0 ± 0.0n = 8	0.41 ± 0.06n = 8	0.24 ± 0.05n = 8	0.34 ± 0.05n = 8	0.0 ± 0.0n = 8	1.9 ± 0.16 *n = 8	0.25 ± 0.06n = 8	0.62 ± 0.07n = 8

BAC—bile acid control diet; TC = taurocholic acid. * *p* < 0.05 vs. WT fed BAC.

## Data Availability

Not applicable.

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
