# Peer review of "The Effects of Taurocholic Acid on Biliary Damage and Liver Fibrosis Are Mediated by Calcitonin-Gene-Related Peptide Signaling"

_cells, 2022, doi:10.3390/cells11091591_

Round 1

Reviewer 1 Report

The authors answered all my remarks.

Author Response

We thank the Reviewer for the constructive comments that helped us to improve the manuscript.

Reviewer 2 Report

The authors responded to most comments. Point 3 was truncated in the copy-paste and point 11 disappeared altogether.

Some of the data have been improved such as the quantification of the mRNA of the genes of interest in cholangiocytes isolated from animals in the different conditions, even though in the case of Tgfb for example transcriptional regulation is less relevant than post-trancriptional regulation. Some photomicrograpghs such as in figure 2 or Figure 5, have been improved.

Howerver, virtually all the other photomicrographs remain of poor quality and poorly presented (lack of arrows…). Fig. 1A is still presented as an increase in CGRP serum levels, even though the differences are not statistically different. No significant increase means no increase (An * has been added by error in Fig. legend). I would delete Fig. 1A altogether. Figure 2 requires a quantification of inflammatory infiltrates.

Most importantly, the link between the different results presented is still missing. Figure 6 (IPA) only shows a network and there is still no justification for the term PKA/ERK1/2/TGF-b1/TGF_bRI/liver angiogenesis axis. This is all the more true that angiogenesis was not investigated at all. The interpretation of bile acid analysis does not add anything either.

Author Response

  1. The authors responded to most comments. Point 3 was truncated in the copy-paste and point 11 disappeared altogether.

We are very sorry for the confusion with the previous 3 and 11 points. We tried to recover them as follow:

Previous point 3: It would be preferable no to mention results in the Methods section, e., Figure 6 mentioned page 4, line 188. Instead, the method of ingenuity pathway analysis that was used to generate figure 6, should be provided in Methods.

We have mentioned and described Figure 6 in Methods section (line 188-191) just because it represents the pathway we studied and, in our opinion, is important to introduce it in the Methods to better understand the reason why we evaluated the link between TC and cAMP/a-CGRP/VEGF/TGF-b1 signaling.

Previous point 11: VEGF1 seems mainly expressed in hepatocytes and comparisons should be made based on quantification.

We agree with the Reviewer. Indeed, VEGF-A is expressed also in hepatocytes, but the intesity of the immunorectivity is higher in biliary cells as visible with the increased magnification of cholangiocytes (Figure 10).

  1. Some of the data have been improved such as the quantification of the mRNA of the genes of interest in cholangiocytes isolated from animals in the different conditions, even though in the case of Tgfb for example transcriptional regulation is less relevant than post-trancriptional regulation. Some photomicrograpghs such as in figure 2 or Figure 5, have been improved. Howerver, virtually all the other photomicrographs remain of poor quality and poorly presented (lack of arrows…). Fig. 1A is still presented as an increase in CGRP serum levels, even though the differences are not statistically different. No significant increase means no increase (An * has been added by error in Fig. legend). I would delete Fig. 1A altogether. Figure 2 requires a quantification of inflammatory infiltrates.

We thank the Reviewer for the critical suggestions. We have attempted to improve the quality of our images and have inserted the arrows in all the figures. In addition, we removed the data regarding serum in Figure 1A and added the quantification of inflammatory infiltrates in Table 1.

  1. Most importantly, the link between the different results presented is still missing. Figure 6 (IPA) only shows a network and there is still no justification for the term PKA/ERK1/2/TGF-b1/TGF_bRI/liver angiogenesis axis. This is all the more true that angiogenesis was not investigated at all. The interpretation of bile acid analysis does not add anything either.

We thank the Reviewer for his comments. Indeed, in the discussion section, we observed that the IPA analysis support our hypothesis that TC may induce the binding of α-CGRP with its CALCRL receptor that regulates cellular events such as fibrogenesis and angiogenesis and it may be through changes in cAMP/ERK1/2/TGF-β1/VEGF signaling axis. Moreover, we agree with the Reviewer that we did not investigate angiogenesis, but just the expression of some important angiogenic factors. For that reason, we changed the term angiogenesis with the expression of angiogenic factors along all the paper. Regarding the comment from the reviewer that the new bile acid species data are not helpful or informative - this is concerning considering the entire paper is related to bile acid regulation of hepatic damage and knowing the specific species adds tremendous value to the previous data. Particularly, there are different studies that demostrated the changes of bile acid determined liver damage (Vitamin E reduces oxidant injury to mitochondria and the hepatotoxicity of taurochenodeoxycholic acid in the rat. PMID: 9428230 - UDCA, NorUDCA, and TUDCA in Liver Diseases: A Review of Their Mechanisms of Action and Clinical Applications. PMID: 31236688) as well as their protective role in cholestatic liver diseases (Ursodeoxycholic Acid Therapy in Treatment of Primary Sclerosing Cholangitis. PMID: 7824879 - Effect of ursodeoxycholic acid on the kinetics of cholic acid and chenodeoxycholic acid in patients with primary sclerosing cholangitis. PMID: 8514251 - High-dose ursodeoxycholic acid for the treatment of primary sclerosing cholangitis. PMID: 17335678). Therefore, in our opinion, these data significantly improve the manuscript.